# Water-Soluble Pristine C_60_ Fullerene Inhibits Liver Alterations Associated with Hepatocellular Carcinoma in Rat

**DOI:** 10.3390/pharmaceutics12090794

**Published:** 2020-08-22

**Authors:** Halyna Kuznietsova, Natalia Dziubenko, Tetiana Herheliuk, Yuriy Prylutskyy, Eric Tauscher, Uwe Ritter, Peter Scharff

**Affiliations:** 1Institute of Biology and Medicine, Taras Shevchenko National University of Kyiv, Volodymyrska str., 64, 01601 Kyiv, Ukraine; n_dziubenko@ukr.net (N.D.); vodolyb@ukr.net (T.H.); prylut@ukr.net (Y.P.); 2Institute of Chemistry and Biotechnology, Technical University of Ilmenau, Weimarer str. 25, 98693 Ilmenau, Germany; eric.taeuscher@tu-ilmenau.de (E.T.); peter.scharff@tu-ilmenau.de (P.S.)

**Keywords:** water-soluble pristine C_60_ fullerene, hepatocellular carcinoma, HepG2 cells, oxidative stress, 5-fluorouracil, DLS measurements

## Abstract

Excessive production of reactive oxygen species is the main cause of hepatocellular carcinoma (HCC) initiation and progression. Water-soluble pristine C_60_ fullerene is a powerful and non-toxic antioxidant, therefore, its effect under rat HCC model and its possible mechanisms were aimed to be discovered. Studies on HepG2 cells (human HCC) demonstrated C_60_ fullerene ability to inhibit cell growth (IC_50_ = 108.2 μmol), to induce apoptosis, to downregulate glucose-6-phosphate dehydrogenase, to upregulate vimentin and p53 expression and to alter HepG2 redox state. If applied to animals experienced HCC in dose of 0.25 mg/kg per day starting at liver cirrhosis stage, C_60_ fullerene improved post-treatment survival similar to reference 5-fluorouracil (31 and 30 compared to 17 weeks) and inhibited metastasis unlike the latter. Furthermore, C_60_ fullerene substantially attenuated liver injury and fibrosis, decreased liver enzymes, and normalized bilirubin and redox markers (elevated by 1.7–7.7 times under HCC). Thus, C_60_ fullerene ability to inhibit HepG2 cell growth and HCC development and metastasis and to improve animal survival was concluded. C_60_ fullerene cytostatic action might be realized through apoptosis induction and glucose-6-phosphate dehydrogenase downregulation in addition to its antioxidant activity.

## 1. Introduction

Liver cancer is one of the most common malignancies and occupies the second place among the causes of cancer deaths in the world. Liver cancer has the fourth-highest incidence for cancer in men following lung, prostate, and stomach cancers, and the seventh-highest incidence in women following breast, lung, cervical, thyroid, colorectal, and stomach cancers. The overall liver cancer mortality reaches 93% for individuals between the ages of 0 and 74 years and has the second-highest death rate for cancers [1]. Hepatocellular carcinoma (HCC)—a malignant neoplasm of hepatocytes’ origin—is one of the most rapidly progressing fatal oncological diseases. It accounts for up to 90% of all primary liver malignant tumors. In 70–80% of cases, malignant transformation is observed in the cirrhotically altered liver [2], and the presence of cirrhosis significantly increases the risk of tumor development [3].

HCC is challenging to diagnose and has an extremely unfavorable prognosis. If adequate therapy is not provided, the survival of HCC patients is about four months after diagnosis. Complete or partial resection and orthotopic or complete liver transplantation remain the ideal treatments for liver cancer. Non-invasive therapy techniques include transarterial techniques (chemo-, radio-, bland embolization) and ablative therapies (radio-, cryo-, microwave ablation) [4]. The standard chemotherapeutic agent recommended against this pathology is a tyrosine kinase inhibitor sorafenib, which can block Raf serine/threonine kinases and receptor tyrosine kinases (vaso-endothelial growth factor receptor and platelet-derived growth factor receptor). US Food and Drug Administration recently approved tyrosine kinase inhibitors having similar activity such as lenvatinib, regorafenib, ramucirumab, and pembrolizumab as first and second-line HCC treatments [3]. However, the above strategies are often ineffective and cause severe complications in many patients.

Liver cancers develop according to a typical pattern: Progressive inflammation followed by fibrous changes and cirrhosis development with subsequent liver cells’ malignant transformation [5]. The specificity of HCC pathogenesis, in particular progressive fibrotic degeneration of liver tissues, causes a frequent lack of response to chemotherapeutics and the rapid development of severe complications. One of the main mechanisms of both fibrous and malignant degeneration of the liver is the excessive production of reactive oxygen species (ROS) and inhibition of the antioxidant defense system, i.e., the development of oxidative stress. The use of natural antioxidants (vitamins, minor amino acids, polyunsaturated fatty acids, and plant extracts) under liver pathologies related to cirrhosis and neoplasia has been addressed in a large number of studies. However, none of the results can be considered as the solution of this task [6]. Artificial compounds with predetermined properties could be the one, as they are not involved in numerous cell metabolic pathways and, thus, allow targeted influence on specific life processes [7]. Biocompatible water-soluble pristine C_60_ fullerene has unique physical-chemical properties that let it scavenge free radicals effectively, i.e., being antioxidant and, thus, to reveal anti-inflammatory and antitumor activities [8,9,10,11].

Furthermore, C_60_ fullerene is nontoxic when used in therapeutic doses and can be accumulated in the liver [12,13], which makes it attractive to affect it. Hence, this nanoparticle is an ideal candidate for the prevention and treatment of liver disease associated with oxidative stress. Therefore, the effect of water-soluble pristine C_60_ fullerene on the liver state of rats experienced HCC, and the possible mechanisms of its effects were aimed to be discovered. To achieve the aim, the following tasks were purposed: (1) To estimate C_60_ fullerene impact on liver morphology and function, animal survival and metastasis, as well as its safety, for revealing the potency of C_60_ fullerene as anti-HCC therapeutic; (2) to assess redox states of the whole liver and liver malignant cells alone for defining if C_60_ fullerene antioxidant capability could contribute to its therapeutic efficacy; (3) to assume the impact of C_60_ fullerene on hepatoma cells viability and expression/activity of essential proteins responsible for malignant cell survival and metastasis (p53, vimentin, glucose-6-phosphate dehydrogenase, lactate dehydrogenase) for clarifying mechanisms of C_60_ fullerene antitumor action.

## 2. Materials and Methods 

### 2.1. Preparation and Characterization of Pristine C_60_ Fullerene Aqueous Colloid Solution (C_60_FAS) 

The highly stable pristine C_60_ fullerene aqueous colloid solution (C_60_FAS; purity >99.5%, concentration 0.15 mg/mL) was prepared according to the original method [14,15]. Briefly, it was based on transferring C_60_ fullerene from toluene to an aqueous phase with the help of ultrasonic treatment. The prepared C_60_FAS is stable within 12 months at temperature +4 °C [16]. 

The dynamic light scattering (DLS) and zeta potential measurements were conducted for ascertaining the hydrodynamic size and electrokinetic potential of the prepared C_60_FAS by use of the Zetasizer Nano-ZS90 (Malvern, Worcestershire, UK) technique. The obtained results were evaluated using the Smoluchowski approximation that is rigorously valid only for spherical-like particles.

### 2.2. In Vitro Assays

The HepG2 cell line was obtained from RE Kavetsky Institute of Experimental Pathology, Oncology and Radiobiology, National Academy of Sciences of Ukraine. The cells were cultured under standard conditions (37 °C, 5% CO_2_, 95% humidity), in Dulbecco’s Modified Eagle Medium (DMEM, Merck, Darmstadt, Germany) contained 10% fetal bovine serum (FBS, Merck, Germany), 2 mM L-glutamine (Merck, Germany) and 40 mg/mL gentamicin (Biopharma, Kyiv, Ukraine). 

#### 2.2.1. Cell Viability and Apoptosis Assays

Cell viability was assessed by 3-(4,5-dimethylthiazol-2-yl)-2,5-diphenyltetrazolium bromide (MTT) assay (Abcam, Cambridge, UK). Cells were seeded on coated glass in 96-well plates with 6 × 10^3^ cells/well density and used in 24 h of cultivation. Then cells were incubated in medium contained 1% PBS and 1, 10, or 100 μg/mL C_60_FAS or 1 and 10 μg/mL Doxorubicin (Dox, Ebewe Pharma, Unterach am Attersee, Austria) as a reference for 48 h, and after that MTT assay was performed according to manufacturer’s kit (Abcam, UK). Values of absorption at different concentrations were normalized to the average control value according to the manufacturer’s protocol. 

Cell death was assessed by Annexin V/Propidium Iodide (AnV/PI) apoptosis assay. Cells were seeded in 6-well plates with 0.7 × 10^6^ cells/well density and incubated in serum-free medium contained 1, 10, or 100 μg/mL C_60_FAS for 48 h, then apoptosis assay was performed according to manufacturer’s kit (Annexin V-FITC Apoptosis Detection Kit I, BD Pharmingen, San Diego, CA, USA). The proportions of alive, dead cells and those in different stages of apoptosis were measured by flow cytometry (FACS Calibur, Becton Dickenson, Franklin Lakes, NJ, USA), the samples were analyzed using CellQuest Software version 5.1 (BD Biosciences, San Jose, CA, USA).

#### 2.2.2. Immunohistochemical Assay

To evaluate vimentin and p53 expressions, the cells were seeded on coated glass in 6-well plates with 2 × 10^4^ cells/cm^2^ density and used in 48 h of cultivation. Cells were incubated in medium contained 10 or 100 μg of C_60_FAS per mL for 48 h. These concentrations were chosen as those demonstrating substantial effect on cell growth and apoptosis. The immunohistochemical assay was performed using primary vimentin and p53 monoclonal antibodies and reagent kits for immunohistochemical visualization according to the protocols provided by manufacturers. We used the following antibodies: anti-vim—clone V9 IS630 (Dako, Carpinteria, CA, USA), anti-p53—DO-1 sc-126 (Santa Cruz Biotechnology, Santa Cruz, CA, USA). Besides, specimens were stained by hematoxylin (Merck, Germany) for nuclei visualization. 

The number of immunopositive cells was counted in 10 random microscopic fields of view for each sample using a standard scale of measurement (object-micrometer) at the same magnification and calculated as a percentage of the total cell number taken for 100%. Light microscope Olympus BX-41 (Olympus Europe GmbH, Munich, Germany) and camera Olympus C-5050 Zoom (Olympus Europe GmbH, Germany) with appropriate software (Olympus DP Soft 3.2, Olympus Europe GmbH, Germany) were used. At least 200 cells were counted. The staining intensity of the cells was assessed using the following scoring: 0—negative, 1—weak; 2—moderate; 3—strong. Histo-score (H-score) was calculated as follows: H-score = 1 × %% weakly stained cells + 2 × %% moderately stained cells + 3 × %% strongly stained cells [17].

#### 2.2.3. Biochemical Assays

HepG2 cells were cultured under standard conditions in medium contained C_60_FAS (10 or 100 μg/mL) for 48 h. Then the cell medium contained C_60_FAS was removed, cells were washed with fresh one (3 times) and were frozen and thawed for destruction; the samples were passed through a syringe needle of medium diameter a few times for homogenization. To assess the impact of C_60_FAS on cell redox state we measured malonic dialdehyde (MDA), protein carbonyl groups (PCG), reduced glutathione (GSH) levels and catalase (CAT), superoxide dismutase (SOD), glutathione-S-transferase (GST), and glutathione peroxidase (GP) activities in the samples. To assess the energy metabolism of the cells, lactate dehydrogenase (LDH), and glucose-6-phosphate dehydrogenase (G6PD) activities were measured. Total protein was estimated using a standard reagent kit (DiagnosticumZrt, Budapest, Hungary). The activity of the enzymes and the contents of low-molecular substances were expressed per mg protein. 

### 2.3. Animal Assay

Experiments were performed using Wistar male rats, which were kept in the vivarium of Taras Shevchenko National University of Kyiv. One-month-old rats with the initial body weight of 120 ± 10 g were distributed randomly in 8 animals per plastic cage on softwood chip bedding; total 80 animals were used in the experiment. Rats were maintained under 12 h light/dark cycle and 50% humidity at 20–22 °C and fed on a standard diet and tap water ad libitum. All experiments were performed in compliance with bioethics principles, legislative norms, and provisions of the European Convention for the Protection of Vertebrate Animals used for Experimental and Other Scientific Purposes (Strasbourg, 1986), General Ethical Principles for Experiments on Animals, adopted by the First National Bioethics Congress (Kyiv, 2001), and approved by an institutional review committee (Ethical Approval Certificate of the project No. 18BF036-01M (MESU grant No. 0118U000244) given by Committee on Bioethics of Educational-Scientific Center “Institute of Biology and Medicine” of Taras Shevchenko National University of Kyiv 15 July 2019).

#### 2.3.1. Design of the Study

HCC was simulated by an initial intraperitoneal injection of N-diethylnitrosamine (DEN, 200 mg/kg) (Merck, Germany) in saline (total volume 0.1 mL per rat). After 2 weeks, subcutaneous injections of carbon tetrachloride (CCl_4_, 1 mL/kg) (Biopharma, Ukraine) in sunflower oil (total volume 0.2–0.7 mL per rat depending on body weight) were started and followed for 20 weeks to stimulate the regeneration process. During the first 10 weeks, we performed 3 injections per week, then 2 injections per week. Changes in liver correspond to advanced fibrosis and cirrhosis (8–12 weeks from the start of the study, i.e., the introduction of DEN), malignant degeneration of liver cells (14–16 weeks from the start of the study) and multiple well-developed liver tumors with metastasis (20–22 weeks) [18,19]. C_60_FAS was administered daily intraperitoneally in a volume corresponded to the dose of C_60_ fullerene 0.25 mg/kg (0.3–0.6 mL per rat depending on body weight), which is considered as safe and effective as evidenced by our previous studies [8,9,20]. Administrations were started at 16th week of the experiment (the stage of cirrhosis and malignant cell degeneration, which is confirmed by our previous research [21]) and followed for 7 weeks to explore C_60_FAS effect on the development of liver tumors and metastasis. We use the common antitumor drug 5-fluorouracil (5FU, Ebewe Pharma, Austria) as a reference. 5FU was injected intraperitoneally weekly in saline at a dose of 15 mg/kg (total volume 0.1 mL per rat) for 7 weeks starting at the 16th week of the experiment. Comparison groups were induced by appropriate solvents instead. Experimental groups (n = 16) were as follows: (1) Control, (2) C_60_FAS, (3) HCC, (4) HCC + C_60_FAS, (5) HCC + 5FU. The scheme of the study is depicted at Figure 1. 

In 24 h after the last dose, halves of the animals from each group (chosen randomly) were sacrificed by inhalation of CO_2_ and subsequent cervical dislocation. Another halves were left without any treatment for the survival test.

#### 2.3.2. Survival Assay

Halves of the animals from each experimental group (n = 8) were observed for the next 32 weeks without any treatment to assess survival; dead animals were dissected, and the metastasis, if any, were registered. In 54 weeks after the start of the experiment, animals that were still alive were sacrificed, and metastasis were recorded. Survival data was assessed by Kaplan–Meier analysis, post-treatment median survival times were calculated, and survival curves were plotted.

#### 2.3.3. Blood Assays

To perform biochemical tests, we harvested the blood immediately after the sacrifice from the femoral vein, left it for 20 min to form a clot, and then centrifuged 10 min at 1500× *g*. Blood serum was collected and used immediately. We determined following serum markers using standard reagent kits (DiagnosticumZrt, Hungary): Alanine aminotransferase (ALT), aspartate aminotransferase (AST), alkaline phosphatase (ALP), LDH, α-amylase activities, contents of the total protein, total (conjugated + non-conjugated) and direct (conjugated) bilirubin, urea, creatinine, and triglycerides.

#### 2.3.4. Histological Assays

Liver, kidney, spleen, and pancreas samples were collected immediately after the sacrifice and fixed in Bouin mixture for 7 days. Then, they were embedded in paraffin, cut into 5 µm-thick slices, and stained with hematoxylin and eosin (H & E, Merck, Germany) according to a standard protocol [22]. We used the following reagents: chloroform, formalin, paraffin, ethanol (Biopharma, Ukraine), hematoxylin, eosin, orange G (Merck, Germany). Possible injuries were examined under the light microscope (Olympus BX-41, Olympus Europe GmbH, Germany) by two independent pathologists who were unaware of the treatment group.

#### 2.3.5. Liver Assays

Liver autopsies were assessed according to the following score [23]: 0 —liver is reddish-brown, with a soft consistency and smooth surface; 1—liver is enlarged, with soft and friable consistency; 2—liver is hyperemic; 3—liver is enlarged with occasional white foci of necrosis; 4—liver is enlarged with multiple white foci of necrosis; 5—hepatic steatosis, manifested by the enlarged, yellow, soft and greasy liver; 6—liver is of average or increased size and mottled red with bile stained areas; 7—liver contains visible nodules; 8—micronodular fibrosis accompanied with the enlarged yellow fatty liver; 9 —macronodular fibrosis and cirrhosis accompanied with the shrunken, brown non-greasy liver; 10—small single granular bodies are presented at explanted liver; 11—large single granular bodies are presented at explanted liver; 12—many small granular bodies are presented at explanted liver; 13—many large granular bodies are presented at explanted liver.

Due to the light microscopy examination, typical or atypical liver histological structure was considered, the shape, structure, and staining of cells, stroma, and gland tissue infiltration by leucocytes and vasculature were assessed and analyzed. The severity of fibrosis was evaluated according to Ishak fibrosis score [24]: 0—no signs of fibrosis; 1—fibrous expansion of some portal areas, short fibrous septs may occur; 2—fibrous expansion of most portal areas, short fibrous septs may occur; 3—fibrous expansion of most portal areas, portal–portal linking septs are occasional; 4—fibrous expansion of most portal areas, portal–portal linking septs are prevalent, portal–central bridging also occurs; 5—fibrous septs are common and well-developed, nodules are occasional; 6—cirrhosis. 

Rests of the livers were washed with saline followed by phosphate-buffered saline (PBS, pH 7.0) containing 1 mM ethylenediaminetetraacetic acid (EDTA, Thermo Fisher Scientific, Waltham, MA, USA) and 0.4 mM phenylmethylsulphonyl fluoride (PMSF, serine proteases inhibitor) (Thermo Fisher Scientific, USA) and rapidly frozen at −70 °C. After being thawed, samples were homogenized in PBS containing 1mM EDTA and 0.4 mM PMSF by hand homogenizer, filtered through four cheesecloth layers and centrifuged (10,000× *g*, 15 min) to sediment the nuclei and mitochondria. Supernatants were harvested and used for analysis immediately—the whole cycle from thawing until analysis was performed on the same day. Contents of MDA, PCG, and GSH, and activities of intracellular SOD, CAT, GP, and total GST, served as liver redox state markers, were measured spectrophotometrically and expressed per mg protein. Total protein was estimated using a standard reagent kit (DiagnosticumZrt, Hungary).

### 2.4. Enzyme Assays

Lipid peroxidation was assessed based on the ability of its final product MDA to react with thiobarbituric acid and to form a colored trimethine complex having a maximum absorption at 532 nm. The test was performed as described at [25], the molar extinction coefficient of 1.56 × 10^5^ M^−1^·cm^−1^ was used. 

Protein oxidation was estimated according to [26] when oxidized amino acid residues react with 2,4-dinitrophenylhydrazine with forming a stable product 2,4-dinitrophenylhydrazone. The extent of 2,4-dinitrophenylhydrazone was determined at 370 nm using a molar extinction coefficient of 2.2 × 10^4^ M^−1^·cm^−1^. 

Content of GSH was assessed using the ability of 5,5′-dithiobis-2-nitrobenzoic acid to oxidize GSH to glutathione disulfide and 5-thio-2-nitrobenzoic acid having a maximum absorption at 412 nm [27]. The content of GSH was quantified using the calibration curve. 

SOD activity was evaluated based on its ability to inhibit nitro blue tetrazolium (NBT) photoreduction, as described by [28]. The reaction was initiated by bright sunshine for 10 min and stopped by adding 1% potassium iodide; absorbance was measured at 540 nm. 1 Unit of SOD activity was defined as the amount of enzyme required to cause 1% inhibition of NBT photoreduction rate. 

CAT activity was estimated based on its ability to split H_2_O_2_ and thus to decrease the forming of colored product in the reaction between H_2_O_2_ and molybdenum salt. The test was performed according to [29], chromogen absorbance was determined at 410 nm, the molar extinction coefficient of 22.2 M^−1^·cm^−1^ was used.

GP activity was evaluated by measuring unconsumed GSH in reaction with t-butyl hydroperoxide using as a substrate [30]. The content of GSH was determined as described above.

Total GST activity was assessed by its ability to transfer GS-group to 1-chloro-2,4-dinitrobenzene. The test was performed according to [31], chromogen absorbance was determined at 340 nm, the molar extinction coefficient of 9600 M^−1^·cm^−1^ was used.

G6PD activity was assessed by its ability to oxidize glucose-6-phosphate to 6-phosphogluconolactone while reducing nicotinamide adenine dinucleotide phosphate (NADP^+^) to NADPH, which is accompanied by an increase of absorbance at 340 nm. The molar extinction coefficient of 6220 M^−1^·cm^−1^ was used [32]. 

LDH activity was assessed by its ability to oxidize nicotinamide adenine dinucleotide H (NADH) while reducing pyruvate to lactate (reverse reaction), which is accompanied by a decrease of absorbance at 340 nm. The molar extinction coefficient of 6220 M^−1^·cm^−1^ was used [33].

The following reagents were used: ice acetic acid, methanol, tret-buthanol, pyridine, sodium tret-butylate, sodium hydrogen phosphate and dihydrogen phosphate, trochloroacetic acid, hydrochloric acid, ethyl-acetate, ethanol, urea, potassium iodide, ammonium molybdate, H_2_O_2_ (Biopharma, Ukraine), tetramethylethylenediamine (Thermo Fisher Scientific, USA), thiobarbituric acid, NBT, riboflavin (Merck, Germany).

### 2.5. Statistical Analysis

Data analysis was performed using SPSS 23.0 software package for Windows. Data distribution was assessed using the Kolmogorov-Smirnov one-sample test. Statistical analysis of the data was performed using a one-way analysis of variance (ANOVA) with the Tukey post hoc test if the data were normally distributed, otherwise the Mann–Whitney test was applied. Survival data were analyzed by the Kaplan–Meyer method, and Breslow criterion was used for comparison of different treatments. The difference was considered statistically significant at *p* < 0.05.

## 3. Results

### 3.1. C_60_FAS Characterization

It is known that aggregation and stability of C_60_ fullerene particles in water affect bioavailability and toxicity to an organism [34,35,36,37]. Therefore, the prepared C_60_FAS (concentration 0.15 mg/mL) was characterized by DLS technique. 

The DLS method has shown that C_60_FAS contains single C_60_ molecules (0.72 nm) as well as their aggregates up to 100 nm that is in a good agreement with the calculations [38]. In addition, the stability of the used C_60_FAS was evaluated by the zeta potential measurement. This value was shown to be −25.4 mV at room temperature. Such a high (by absolute value) zeta potential for the tested C_60_FAS indicates high stability (low tendency for nanoparticle aggregation over time) and its suitability for further biological research.

### 3.2. HepG2 Assays

Studying the effect of C_60_FAS on HepG2 cells (human HCC), we observed the oppression of cell viability by C_60_ fullerene. We determined IC_50_ as 77.9 μg/mL (corresponds to 108.2 μmol) (Figure 2), which allows us to conclude its moderate toxicity [39,40]. Indeed, 5FU IC_50_ for these cells is equal to 323 μmol, whereas Dox one—only 1.1 μmol [41]. Therefore, we might assume that C_60_ fullerene is a rather cytostatic agent for HepG2 cells than cytotoxic one. Further, we determined if apoptosis contributes to inhibition of cell survival by C_60_FAS. We observed dose-dependent increase of early- and late-apoptotic cell pools and no changes in necrotic cell pool after incubation with C_60_FAS (Figure 2). Thus, we could conclude the ability of C_60_FAS to induce apoptosis in HepG2 cells if applied in high doses (>10 µg/mL).

Studying the effect of C_60_FAS on vimentin and p53 expression in HepG2 cells, we found that C_60_ fullerene stimulated the expression of both (Figure 3). Thus, H-scores for vimentin and p53 increased in a dose-dependent manner by 93–183% and 73–86%, respectively (Table 1).

We observed that C_60_ fullerene inhibited G6PD activity in a dose-dependent manner up to almost entirely blocking through the cultivation at maximum concentration (100 µg/mL). LDH activity increased (up to 64%) at the same time. Changes of HepG2 redox state markers were ambiguous. There was an increase in SOD activity up to 93% and a decrease in PCG down to 94% (suggesting the oxidative stress diminution) simultaneously with CAT, GST, GP and GSH suppression (by 22–46%) and MDA elevation up to 6.5 times (signs of oxidative stress) (Table 1). Moreover, changes of SOD, MDA and LDH were not dose-dependent, probably through different mechanisms of action of different (one order of magnitude) doses of C_60_ fullerene. 

### 3.3. Survival Assay

An analysis of post-treatment survival demonstrated a 76% and 82% increase in the median survival of animals exposed to 5FU and C_60_FAS, respectively, compared to non-treated ones (Figure 4, Table 2).

Moreover, analyzing the autopsies and slides of the pancreas, we observed no atypical cells in the pancreas at the 22nd and 54th week in the C_60_FAS group but well-developed metastasis at the 54th week in 5FU group. In contrast, non-treated animals demonstrated neoplastic cells aggregates and even well-developed tumors at the 22nd week and massive metastasis at the 54th one (Appendix A).

### 3.4. Liver Assays

Healthy animals received C_60_FAS for 7 weeks did not reveal any changes in overall looking, liver, kidney, pancreas, and spleen morphologies and blood serum markers except elevated triglycerides (up to 2.2 times) and depressed α-amylase activity (down to 3.7 times), indicating possible pancreatic dysfunction (Table 2, Appendix A). The liver redox state was not altered as well (Table 3).

In HCC-experienced animals received no treatment, multiple light nodes of different size under the cirrhotic background were observed in the liver (Figure 5). The livers were dark, suggesting the blood stasis. Microscopy studies revealed well-developed cirrhosis and foci of hypertrophic hepatocytes with loss of cytoplasm granularity, which might suggest focal nodular hyperplasia or hepatocellular adenoma [42]. We also observed the loci of high-developed HCC (probably), hepatocytes’ cytoplasm dystrophic alteration (eosinophilic alteration), enlarged nuclei with decondensed chromatin (suggesting increased activity) (Figure 5). Dilated and overflowed vessels could evidence portal hypertension. Significant elevation of conjugated and non-conjugated bilirubin (to 2.7 and 2.17 times, respectively), total protein (by 33%), AST (by 74%) and GGT (to 4.8 times) in blood serum was observed, other markers remained at the control level (Table 2). The massive replacement of liver parenchyma by fibrous and tumor tissue might explain the absence of ALT, ALP, and LDH elevation, due to the reduction of the quantity of fully functioning hepatocytes and, accordingly, the values of the above enzymes within the normal range. Light microscopy examination of the pancreas revealed edema, blood vessels overflow, tissue fibrous replacement in some autopsies. Kidney and spleen were likely unaltered (Appendix A). 

Liver tissue MDA and PCG tended up (to 3.7 and 3.9 times, respectively), suggesting oxidative stress. However, antioxidant enzymes’ activity and GSH also increased (to 1.9–7.7 times) (Table 3), which may be a sign of tumor adaptation to oxidative stress and is quite typical for HCC [43,44].

The livers of HCC-experienced animals treated with 5FU were substantially consistent with those of non-treated ones and demonstrated multiple light nodes of different sizes and cirrhotic features. The color of the livers was light (yellowish) or dark (Figure 5). The microscopic studies, however, revealed significant diminution of cirrhotic alteration (by 40% according to Ischak scoring): portal–portal septs were thin with unaltered liver tissue architecture between them, often fibrous expansion was limited by portal areas only. Nevertheless, hepatocytes eosinophilic alteration, the signs of portal hypertension, focal nodular hyperplasia and adenoma also occurred (Figure 5). Serum conjugated and non-conjugated bilirubin levels in HCC + 5FU-rats were lower compared to HCC-animals with no treatment (by 43% and 32%, respectively) remaining, however, higher than those in control (by 56% and 49%, respectively). AST and GGT retained elevated, α-amylase activity decreased (by 55%) (Table 2). Liver PCG and MDA in HCC + 5FU-rats were diminished compared to non-treated HCC-rats (by 41% and 53%, respectively, retaining elevated compared to control by 130% and 72%), as well as antioxidant defense enzymes and GSH (by 36–89% compared to non-treated animals, changed insignificantly compared to control) (Table 3). Observed changes may indicate a normalizing of liver redox state after 5FU treatment, which is probably related to the overall improvement of the organ. Pancreas, kidney, and spleen were without significant alterations, as evidenced by histological examination (Appendix A).

The livers of C_60_FAS-treated rats experienced HCC were enlarged, yellowish, had smooth margins. The pieces of evidence of fibrosis manifested by single or multiple soft nodes were observed (Figure 5). Microscopy examination demonstrated substantial inhibiting the cirrhotic alteration (by 35% according to Ischak scoring): Portal–portal septs were thin with unaltered liver tissue architecture between them, often fibrous expansion was limited by portal areas only without connecting bridges. However, foci of eosinophilic and basophilic alteration and the signs of portal hypertension also took place (Figure 5). Serum conjugated and non-conjugated bilirubin in HCC + C_60_FAS-rats were leveled (unlike 5FU group). AST remained elevated at HCC-animals’ level. HCC + C_60_FAS-rats’ GGT was also higher than that in control (to 4 times); however, it tended down compared to HCC-group. We detected serum triglycerides growth up compared to both control and HCC-group (to 2.8 and 3.16 times, respectively) and α-amylase’s fall as well (by 90%) in HCC animals received C_60_FAS (Table 2). The latter may be related to the adverse effects of C_60_ on the pancreas and carbohydrate metabolism disorders. However, no significant alterations were observed in pancreatic, renal, and spleen structures (Appendix A). HCC + C_60_FAS animals, as well as the HCC + 5FU ones, demonstrated MDA and PCG decrease (by 74% and 27%, respectively) compared to non-treated HCC-rats. MDA was normalized, whereas PCG remained elevated to 2.8 times compared to control. Antioxidant defense markers in HCC + C_60_FAS-rats were leveled (and lowered compared to HCC-group by 57–82%) (Table 3). Hence, the C_60_FAS administration contributed to the normalization of the liver redox state. 

## 4. Discussion

We have shown that C_60_ fullerene exerts a probable cytostatic effect on HepG2 cells, exacerbates the pool of apoptotic cells in a dose-dependent manner and upregulates the expression of pro-apoptotic protein p53. As p53 is one of the main pro-apoptotic proteins, and C_60_ fullerene has been shown to induce p53 expression [45] we assume that p53 overexpression might contribute to C_60_ fullerene cytostatic effect on HepG2 cells through apoptosis induction. However, if applied in doses less than 10–20 µg/mL (depending on cell line) C_60_ fullerene didn’t increase the number of apoptotic and pre-apoptotic cells [46], being, therefore, relatively low-toxic. Low cytotoxicity of C_60_ fullerene against normal and neoplastic cells was also demonstrated in [13,47]. Thus, C_60_ fullerene did not affect leukemic cell viability when applying at 3.6–144 µg/mL (CCRF-CEM, Jurkat, Molt-16 cells), IC_50_ for HEK293 cells was equal to 383.4 µg/mL. 

Surprisingly, despite of fibrosis and HCC development and metastasis diminution in vivo, we observed enhanced expression of vimentin in HepG2 cells caused by C_60_FAS. It is known that upregulation of vimentin is a marker of epithelial-mesenchymal transition, which indicates an increase in the ability of tumor cells to migrate and invade and, therefore, tumor ability to metastasize [48]. Moreover, activated hepatic stellate cells under liver fibrosis are characterized by vimentin overexpression, indicating their mesenchymal phenotype [49]. At the same time, HepG2 cells if exposed to lipopolysaccharide have been shown to reveal enhanced vimentin expression and cleavage, and amplified apoptosis rate [50]. Furthermore, enhanced expression of vimentin in ultraviolet-irradiated skin fibroblasts followed by increased cell apoptosis was shown in [51], and caspase-triggered proteolysis of vimentin was proposed as the stimuli of that. Additionally, in the study [52] vimentin involvement in tumor necrosis factor-α-induced cell apoptosis was described. Moreover, Li et al. [53] constructed HepG2 cells highly expressing vimentin and observed reduced cell viability and invasiveness. As they assessed the proportion of alive cells only, it is possible that enhanced cell apoptosis contributes to reduced cell count. We found that C_60_FAS induced apoptosis in HepG2 cells, so we can suppose that vimentin upregulation could be related with that. As C_60_FAS if applied under HCC inhibited tumor development and metastasis, we could speculate that C_60_ fullerene is unlikely to induce epithelial-mesenchymal transition in vivo (at least if applied in tested dose due to 7 weeks) but rather causes apoptosis of tumor cells. Moreover, taken into consideration that C_60_ fullerene inhibits the progression of liver fibrosis, we could assume that it unlikely contributes to activation of hepatic stellate cells but might inhibit their viability instead. Furthermore, the ability of C_60_ fullerene to inhibit liver fibrosis and cirrhosis was demonstrated in our previous results [20,21]. Meanwhile, the obtained in vitro results are controversial and require further investigation.

According to our data, C_60_ fullerene inhibited G6PD activity in HepG2 cells down to almost entirely blocking in a dose-dependent manner. G6PD is the first and key rate-controlling enzyme of the pentose-phosphate pathway (PPP) which provides ribose and NADPH that support biosynthesis and antioxidant defense [54]. Thus, G6PD is highly expressed in tumor cells compared to normal ones. Downregulation of this enzyme leads to the inhibition of cell proliferation and its apoptotic death, especially under oxidative stress conditions. That is why the oppression of G6PD is one of the potential strategies for tumor growth inhibition and overcoming drug resistance [55]. Since G6PD inhibition is the most striking event compared to other observed biochemical changes in HepG2 cells after cultivation with C_60_FAS, we can assume that redox balance alterations, namely MDA increase and CAT, GST, GP, and GSH decrease, might be a consequence of the HepG2 cells’ redox homeostasis disturbance. However, we assume that the ability of C_60_ fullerene to inhibit G6PD activity contributes a lot to the cytostatic action of the nanoparticle unlikely through the oxidative stress induction but instead through PPP product deprivation. Indeed, if we compare the effects of C_60_FAS on the redox state of HepG2 cells and HCC-animals and take into account that slight oxidative stress is typical for tumor cells, we observe the similar changes in redox markers. In both cases, products of lipid and protein peroxidation were decreased as well as the activities of antioxidant defense enzymes and GSH content. The latter becomes clear when we recall that the antioxidant defense system in cancer cells is overactivated in response to the stress [43,44], which is in line with our in vivo results. So, both in vitro and in vivo C_60_FAS likely reverses the redox state to “normal” and non-tumorigenic state: Depresses lipid peroxidation and diminishes stress-induced overactivation of antioxidant enzymes. However, despite SOD activity was reversed to control in in vivo study, in vitro results demonstrated significant elevation of the enzyme. This could be explained by probable different effects of C_60_ fullerene on SOD isozymes. Thus, Cu/ZnSOD is predominantly localized in the cytosol and is essential for the regulation of cytoplasmic H_2_O_2_. In contrast, MnSOD is a mitochondrial enzyme and is required for maintaining mitochondrial integrity and functions and for sustaining a metabolic switch from mitochondrial respiration to glycolysis (Warburg effect) [56]. In [57], the authors explained the observed ability of C_60_FAS to diminish MnSOD overactivation through stabilization of the mitochondrial membrane and thus preventing MnSOD “leakage” from mitochondria to cytosol. As we assessed total SOD activity in HepG2 cells’ homogenate and HCC livers’ cytosolic fraction (after separation the most of mitochondria), we might guess the observed total SOD diminution in HCC samples to be the result of MnSOD preservation in mitochondria and, thus, its removal from the tested supernatants. However, this topic needs further research.

We showed attenuation of liver injury due to C_60_FAS action and significant suppression of its fibrotic degeneration on HCC model in vivo and no signs of toxicity against other vital organs (kidney, spleen). However, C_60_FAS might affect pancreatic function, as evidenced by blood serum α-amylase and triglycerides. It is worth noting that non-alcoholic fatty liver disease (NAFLD) often accompanies pancreatic insufficiency; moreover, low serum α-amylase is significantly associated with NAFLD regardless of concomitant pathologies [58,59]. So, the pancreas could be considered as an organ, most sensitive to C_60_ fullerene, which is in line with our previous studies [20], and therefore has to be thoroughly analyzed while investigating the bioactivity of this nanoparticle.

Chronic inflammation accompanied by excessive ROS production and antioxidant defense system depression and, consequently, oxidative stress is known to be the basis of the vast majority of liver diseases, including fibrosis, cirrhosis, and neoplasia [6,60]. Indeed, many antioxidants have been shown to suppress both chronic liver inflammation and its consequences, including fibrosis, cirrhosis, and neoplasia [61]. They could realize their activity through direct ROS scavenging or modulating ROS-dependent signaling pathways including ROS prevention and amplification [62], inhibition of NADPH oxidases [63], Wnt [64], NF-κB [65], and cytokine signaling [66,67], activation of antioxidant defense system [68]. However, often it is difficult to determine the specific mechanism of action of the chemical because, in most cases, the action of antioxidants is complex and involves both ROS scavenging and modulation of one or more ROS-dependent signaling pathways. Furthermore, frequently anti-fibrotic or anti-inflammatory activity of the substance is not directly related to the suppression of ROS-dependent signaling. However, it affects the cell redox state because of a close relationship with numerous other signalings. For example, growth factor receptor inhibitors have been shown to release anti-inflammatory and anti-fibrotic activity also exhibit antioxidant one [69,70]. Although there are no proven anti-fibrotic therapeutics, numerous studies are underway to identify new ones and to establish anti-fibrotic activity for popular treatments of liver diseases accompanied by its fibrous degeneration [71]. Lots of studies have revealed the prophylactic effect of natural antioxidants against HCC [72,73]. However, simple free radical scavengers demonstrated a lack of efficacy, probably because of cell compensatory systems and a complex network of signaling pathways designed to ensure tumor cells’ survival and proliferation. Therefore, attention should be focused on compounds affecting pro-inflammatory, apoptotic, survival, and proliferative signaling pathways in addition to direct antioxidant activity [72]. Indeed, C_60_ fullerene having been shown to possess potent antioxidant activity because of free radical scavenging [74,75], also could modulate p53 expression and G6PD activity and induce apoptosis if applied in relatively high doses. Moreover, it was shown that C_60_ fullerene is able to interact with epithelial growth factor receptor (EGFR) and fibroblast growth factor receptor (FGFR) [21], to affect the expression of extracellular matrix proteins pan-cytokeratins [20] and to modulate the immune response [75,76]. Herein, we could suggest the multiplicity of the mechanisms of C_60_ fullerene action, which could contribute (in our opinion) to its substantial anti-inflammatory and anti-fibrotic effect because of multiplicity of cellular targets and, therefore, ways of action. Furthermore, obtained results of the anti-inflammatory, anti-fibrotic, and hepatoprotective effects of C_60_FAS correspond to our previous studies [8,9,20,21,77]. It is important to note that C_60_FAS is a relatively safe substance: we did not observe any alterations of vital organs (liver, kidney, spleen, pancreas) or other signs of acute or subacute toxicity in animals receiving C_60_FAS for 7 weeks. Moreover, the maximum tolerated dose of C_60_ fullerene was found to be 721 mg/kg for intraperitoneal administration to mice [13], and any significant violations appeared at the dose of C_60_FAS 150 mg/kg, which is significantly lower than that used in our experiment. 

The obtained data on anti-inflammatory, anti-fibrotic, antitumor, and hepatoprotective action of C_60_ fullerene allow us to offer this nanoparticle as promising therapeutic for correction of liver malignancies. The mechanisms of C_60_ fullerene biological activity presumably are not limited by its ability to scavenge free radicals. However, they might include the cytostatic activity against neoplastic cells, the ability to downregulate G6PD, to affect the expression of extracellular matrix proteins and p53 and to interact with EGFR and FGFR.

## 5. Conclusions

Thus, the ability of C_60_ fullerene if administered as an aqueous colloid solution to inhibit fibrotic and cancerous degeneration and metastasis and to improve animal survival on DEN + CCl_4_-induced rat HCC model has been demonstrated. This C_60_ fullerene action might be realized through its antioxidant properties, i.e., the ability to normalize liver redox state, and cytostatic activity against malignant cells. The ability of C_60_ fullerene to upregulate the expression of p53 and to suppress G6PD could contribute to the latter.

## Figures and Tables

**Figure 1 pharmaceutics-12-00794-f001:**
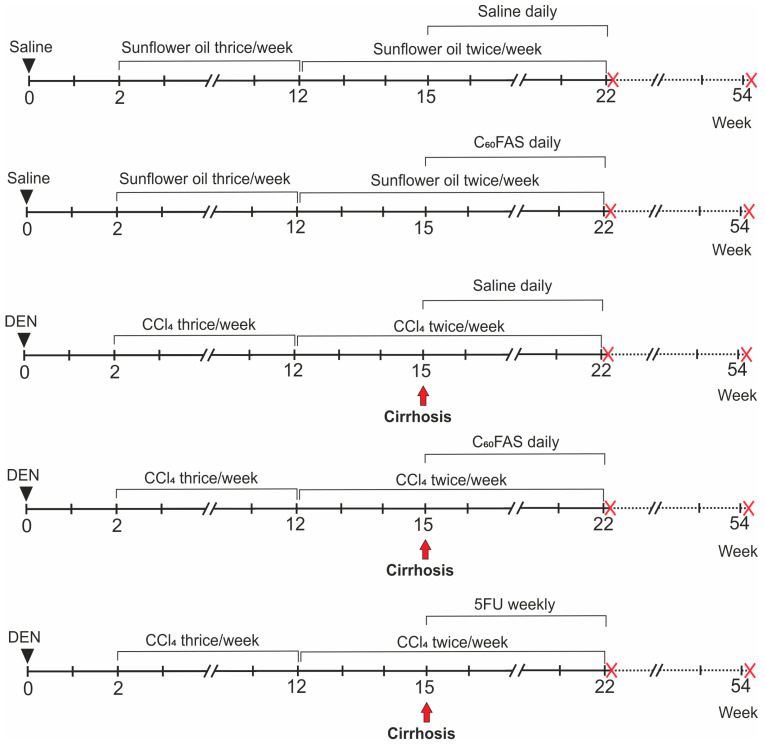
The scheme of the experiment. Experimental groups were as follows (from top to bottom): Control (n = 16); C_60_FAS (n = 16); HCC (n = 16); HCC + C_60_FAS (n = 16); HCC + 5FU (n = 16). Doses and solvents: DEN—200 mg/kg in saline (total volume 0.1 mL), CCl_4_—1 mL/kg in sunflower oil (total volume 0.2–0.7 mL depending on body weight), C_60_FAS—0.25 mg/kg (0.3–0.6 mL depending on body weight), 5FU—15 mg/kg in saline (total volume 0.1 mL), saline—0.1 mL (instead of DEN and 5FU) or 0.3–0.6 mL (instead of C_60_FAS), sunflower oil—0.2–0.7 mL (instead of CCl_4_). Halves of each group were sacrificed in 22 weeks from the start of experiment (the first crosses), other halves were left for survival, and animals that were still alive were sacrificed in 54 weeks from the start of experiment (the second crosses).

**Figure 2 pharmaceutics-12-00794-f002:**
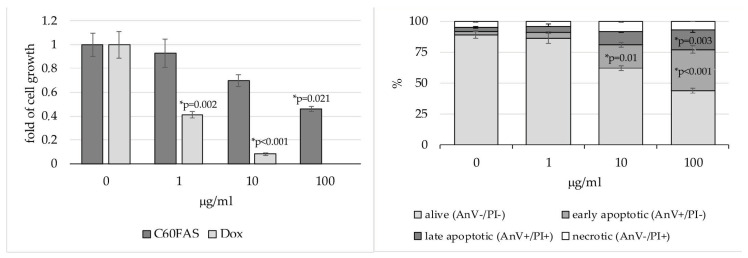
HepG2 cell viability after 48 h cultivation in medium contained C_60_FAS or Dox as a reference (**left panel**) and distribution according to alive, apoptotic and necrotic cell pools after 48 h cultivation in medium contained C_60_FAS (**right panel**); * compared to respective control (*p*-values provided if *p* < 0.1).

**Figure 3 pharmaceutics-12-00794-f003:**
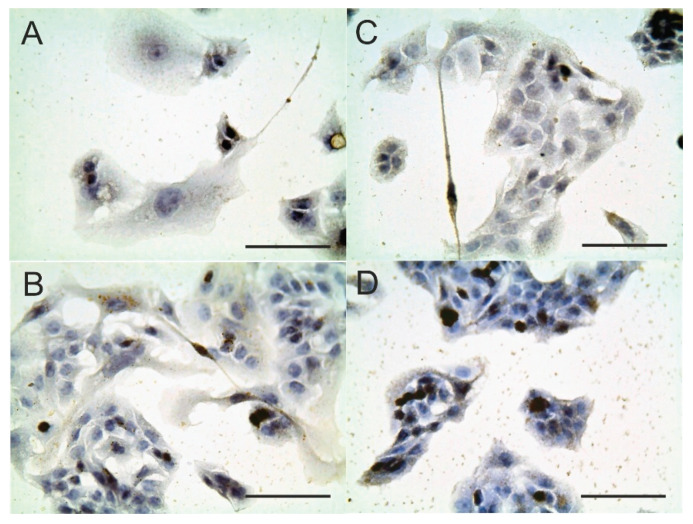
Vimentin and p53 expression in HepG2 cells after 48 h cultivation in medium contained 10 µg/mL C_60_FAS: (**A**): vimentin expression, control; (**B**): vimentin expression, C_60_FAS; (**C**): p53 expression, control; (**D**): p53 expression, C_60_FAS; hematoxylin, ×400, scale 100 µm.

**Figure 4 pharmaceutics-12-00794-f004:**
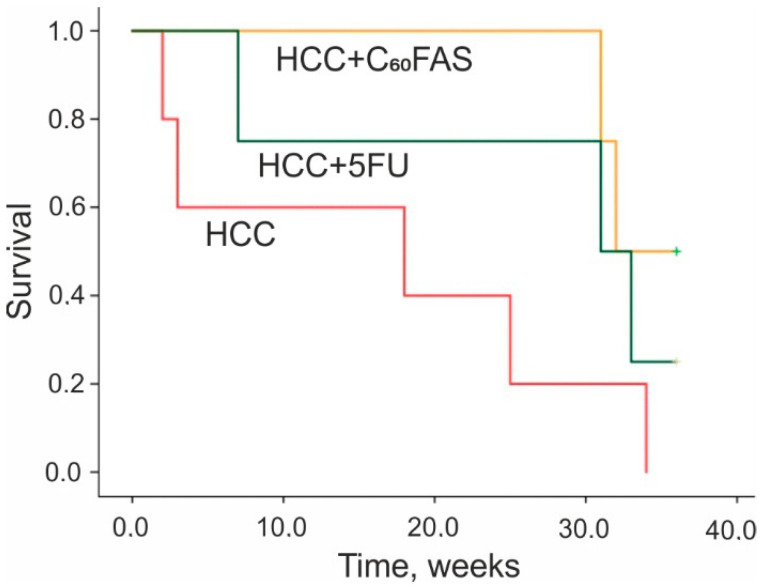
Kaplan–Meyer post-treatment survival curves for HCC-experienced animals treated by 5FU and C_60_FAS.

**Figure 5 pharmaceutics-12-00794-f005:**
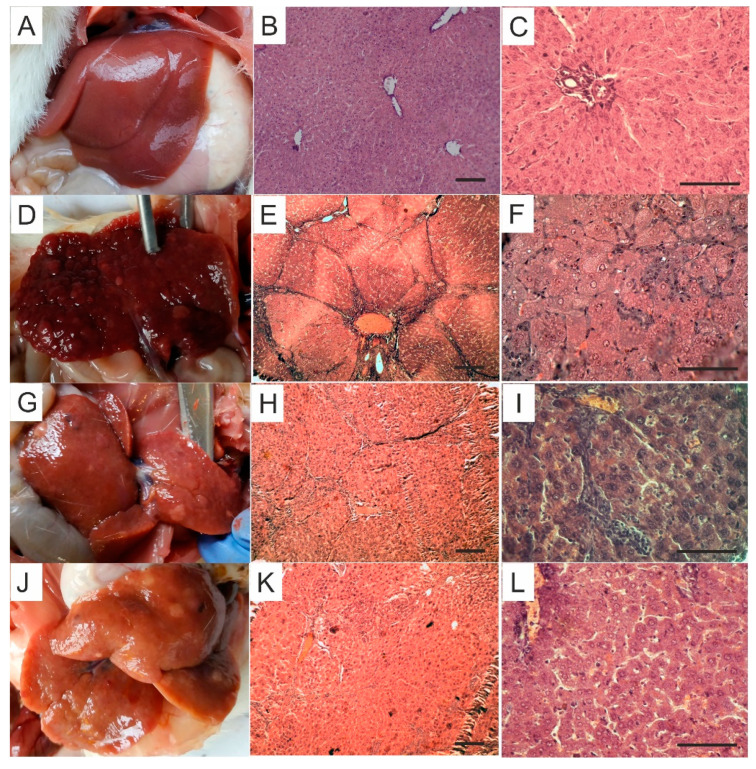
Representative rat livers at the time of the sacrifice (**A**,**D**,**G**,**J**) and H&E staining of liver tissue, magnification ×100 (**B**,**E**,**H**,**K**), scale 200 µm, and ×400 (**C**,**F**,**I**,**L**), scale 100 µm: (**A**–**C**)—control; (**D**–**F**)—HCC; (**G**–**I**)—HCC + 5FU; (**J**–**L**)—HCC + C_60_FAS.

**Table 1 pharmaceutics-12-00794-t001:** H-score, redox and energy state markers of HepG2 cells after 48 h cultivation in medium contained C_60_FAS (M ± SEM).

Parameters	C_60_FAS 0 µg/mL (Control)	C_60_FAS 10 µg/mL	C_60_FAS 100 µg/mL
Vim H-score	13.0 ± 1.3	25.1 ± 3.4 * *p* = 0.012	36.8 ± 5.1 * *p* = 0.003
p53 H-score	18.2 ± 2.2	31.5 ± 4.1 * *p* = 0.02	33.9 ± 4.8 * *p* = 0.006
CAT, µmol/mg prot per min	1551.8 ± 27.1	1458.9 ± 55.3	1151.3 ± 22.7 * *p* < 0.001
SOD, U/mg prot	14.68 ± 1.14	28.4 ± 0.41 * *p* < 0.001	19.17 ± 0.62 * *p* = 0.016
GSH, nmol/mg prot	1.43 ± 0.17	1.31 ± 0.09	0.89 ± 0.002 * *p* = 0.007
GP, nmol/mg prot per min	0.29 ± 0.01	0.31 ± 0.02	0.23 ± 0.01 * *p* = 0.063
GST, nmol/mg prot per min	0.49 ± 0.08	0.54 ± 0.02	0.26 ± 0.01 * *p* = 0.013
PCG, nmol/mg prot	29.27 ± 7.14	23.81 ± 10.94	1.56 ± 0.62 * *p* = 0.01
MDA, nmol/mg prot	23.32 ± 6.47	151.91 ± 39.98 * *p* = 0.008	58.53 ± 6.59
LDH, µmol/mg prot per min	1.48 ± 0.14	2.42 ± 0.3 * *p* = 0.001	1.79 ± 0.08
G6PD, µmol/mg prot per min	6.99 ± 0.24	3.75 ± 0.45 * *p* < 0.001	0.06 ± 0.04 * *p* < 0.001

* compared to control (C_60_FAS 0 µg/mL) (*p*-values provided if *p* < 0.1).

**Table 2 pharmaceutics-12-00794-t002:** Serum biochemical markers of HCC-rats treated by 5FU and C_60_FAS (M ± SEM).

Markers	Control	C_60_FAS	HCC	HCC + 5FU	HCC + C_60_FAS
Direct (conjugated) bilirubin, µmol/L	4.3 ± 0.35	3.95 ± 0.28	11.65 ± 0.33 * *p* < 0.001	6.70 ± 0.85 * *p* = 0.052 ^#^ *p* = 0.001	4.55 ± 0.82 ^#^ *p* < 0.001
Total bilirubin, µmol/L	13.04 ± 1.54	16.43 ± 1.8	30.47 ± 0.87 * *p* < 0.001	19.56 ± 2.22 * *p* = 0.07 ^#^ *p* = 0.007	13.31 ± 1.74 ^#^ *p* < 0.001
Non-conjugated bilirubin, µmol/L	8.64 ± 1.59	12.49 ± 1.65	18.82 ± 0.6 * *p* = 0.003	12.86 ± 1.84 ^#^ *p* = 0.097	8.76 ± 1.18 ^#^ *p* = 0.005
Urea, mmol/L	2.13 ± 0.59	3.46 ± 0.31	2.51 ± 0.39	3.38 ± 0.29	3.63 ± 0.53
Total protein, g/L	45.51 ± 1.87	47.0 ± 1.34	60.63 ± 1.36 * *p* = 0.015	41.24 ± 4.98 ^#^ *p* = 0.005	50.4 ± 2.03
ALP, µmol/L per min	333.31 ± 46.05	335.05 ± 32.7	405.05 ± 2.89	280.42 ± 24.06	303.16 ± 41.86
Triglycerides, mmol/L	2.55 ± 0.32	5.67 ± 1.96 * *p* = 0.067	2.25 ± 0.03	1.87 ± 0.19	7.13 ± 1.07 * *p* = 0.003 ^#^ *p* = 0.002
LDH, IU/L	165.65 ± 24.39	151.39 ± 35.12	180.89 ± 5.5	188.36 ± 34.45	228.43 ± 34.72
α-amylase, g/L per h	106.34 ± 7.22	28.42 ± 16.94 * *p* = 0.001	98.2 ± 10.09	47.14 ± 15.34 * *p* = 0.004 ^#^ *p* = 0.031	9.48 ± 4.09 * *p* < 0.001 ^#^ *p* = 0.001
ALT, µmol/mL per h	1.59 ± 0.1	1.25 ± 0.21	2.06 ± 0.36	1.83 ± 0.33	1.74 ± 0.15
AST, µmol/mL per h	1.71 ± 0.05	1.6 ± 0.11	2.97 ± 0.19 * *p* < 0.001	2.54 ± 0.14 * *p* = 0.001	2.63 ± 0.11 * *p* < 0.001
GGT, IU/L	1.52 ± 0.11	1.79 ± 1.19	7.28 ± 0.38 * *p* = 0.03	7.35 ± 2.12 * *p* = 0.018	6.18 ± 0.4 * *p* = 0.06

* compared to control (*p*-values provided if *p* < 0.1); ^#^ compared to HCC-group (*p*-values provided if *p* < 0.1).

**Table 3 pharmaceutics-12-00794-t003:** Median survival, liver injury and redox state markers of HCC-rats treated by 5FU and C_60_FAS (50 [25;75] percentiles).

Markers	Control	C_60_FAS	HCC	HCC + 5FU	HCC + C_60_FAS
Median survival	N/A	N/A	17 [23;3]	30 [32;7]	31 [35;30] ^#^ *p* = 0.047
Macro-scoring	0	0	10 [10;11] * *p* < 0.001	10 [5;12] * *p* < 0.001	7 [4.75;10] * *p* < 0.001
Ischak scoring	0	0	5 [5;5] * *p* < 0.001	3 [2.25;3.38] *p < 0.001 ^#^ *p* = 0.031	3.25 [2.25;3.88] * *p* < 0.001 ^#^ *p* = 0.032
CAT, mmol/mg prot per min	1.02 [0.76;1.28]	1.35 [0.83;2.12]	1.93 [1.02;2.01]	0.84 [0.72;1.06] ^#^ *p* = 0.077	0.81 [0.72;0.95] ^#^ *p* = 0.034
SOD, U/mg prot	34.26 [27.87;40.75]	71.34 [35.87;88.01] * *p* = 0.095	266.0 [186.48;270.0] * *p* = 0.025	29.59 [22.79;34.09] ^#^ *p* = 0.034	48.71 [35.74;52.31] ^#^ *p* = 0.034
GSH, nmol/mg prot	0.16 [0.12;0.23]	0.2 [0.11;0.27]	0.24 [0.21;0.3] * *p* = 0.099	0.16 [0.11;0.31]	0.23 [0.13;0.28]
GP, nmol/mg prot per min	0.26 [0.15;0.73]	0.27 [0.2;0.35]	0.35 [0.34;0.36]	0.16 [0.1;0.28] ^#^ *p* = 0.032	0.29 [0.17;0.35] ^#^ *p* = 0.067
GST, nmol/mg prot per min	53.66 [49.19;97.8]	88.78 [33.04;185.82]	125.12 [84.03;130.34]	80.69 [53.29;110.82]	50.08 [33.53;71.6] ^#^ *p* = 0.034
PCG, nmol/mg prot	49.5 [27.29;71.72]	68.76 [41.68;150.13]	192.83 [51.87;215.03]	113.7 [54.92;150.46]	141.11 [112.06;150.92] * *p* = 0.014
MDA, nmol/mg prot	76.67 [44.98;100.0]	160.02 [60.11; 176.67] * *p* = 0.074	281.0 [96.64;310.11] * *p* = 0.072	130.86 [85.05;193.34] * *p* = 0.086	71.71 [65.0;115.83] ^#^ *p* = 0.077

N/A—not applicable; * compared to control (*p*-values provided if *p* < 0.1); ^#^ compared to HCC-group (*p*-values provided if *p* < 0.1).

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
