# Peer review of "Water-Soluble Pristine C_60_ Fullerene Inhibits Liver Alterations Associated with Hepatocellular Carcinoma in Rat"

_pharmaceutics, 2020, doi:10.3390/pharmaceutics12090794_

Round 1
Reviewer 1 Report
The manuscript by Kuznietsovaet al. describes the potential usefulness of C60 fullerene for the treatment of hepatocellular carcinoma (HCC). The authors hypothesize that the strong antioxidant properties of C60 fullerene could impact the redox state of liver tumor cells. To evaluate this possibility, they performed in vitro experiments to assess the effect of C60 fullerene on HepG2 proliferation, protein expression and redox state. Then, they perform a series of in vivo experiments in rats with DEN induced HCC to test the in vivo antitumoral properties of C60 fullerene. Here they show that C60 fullerene attenuated liver injury, fibrosis and metastasis.
Although the results concerning the detailed study on the redox state of their models obtained are of potential interest, the significance of the article is severely dampened by lack of experimental evidence to support most of the authors claims during the discussion section. These issues are summarized in the section “comments to the author”.
Major points
- The approaches to evaluate the mechanism of action of C60 fullerene in HepG2 cells are clearly insufficient. First of all, if the intention is to evaluate and quantify changes in the expression of proteins in treated HepG2 cells, a much more direct and easy approach is the western blot. In addition, the quality of the immunocytochemical analyses performed in these cells is poor.
Moreover, to properly study this issue, authors should have performed analysis to evaluate the migratory capacity of the cells as well as the effects of C60 fullerene in the induction of apoptosis or the reduction of cell proliferation. The analysis of the expression of two proteins in one cell line can not account for the conclusions extracted from these experiments.
More importantly, one fails to understand the rationale of the conclusions achieved by the authors in the discussion section (lines 351-362). The authors linked an increase of vimentin expression in HepG2 cells (a protein which has been extensively characterized as a protein involved in the EMT process that results in the increment of the migratory and invasive capacities of the cells, which finally lead to the metastatic spreading of tumors) with the resolution of fibrosis in their animals by metalloproteinase action, and that is without measuring anything related to these proteins (i.e. activity and expression) in any of the models used in this study.
- It is interesting, as well as unsettling, that around 40% of the rats die after few weeks (3-4 weeks according to Figure 3) of treatment, when advanced fibrosis and cirrhosis is achieved 8-12 weeks after starting the treatment. What is the explanation for this?
- It is also curious that authors choose 5-FU as standard drug to compare the efficacy of C60 fullerene when the current first line approved treatment for HCC is the tyrosine kinase inhibitor sorafenib, and more recently the focus of attention in this area is the application of immunotherapy. What is the explanation for this?
- Another overachieving explanation for the results of this study is found in discussion section (lines 394-405). Authors stablished a link around NRF2 based on a couple of references, generating even Figure 5 to explain these interactions. However, NRF2 or anything related with this protein is measured or evaluated in any of the models used in this study. Why authors did not determine the expression/localization of NRF2 in HepG2 cells of livers from rats by the same immunocytochemical assays showed in Figure 2?
- Another inexplicable sentence is found in lines 448-450: “Therefore, we could suggest that the mechanisms of C60 fullerene action are complex, and this fact, in our opinion, explains its significant anti-inflammatory and anti-fibrotic effect”. What is the meaning of this?
- Finally, the manuscript should be corrected by an English native speaker since multiple grammar errors are found through the manuscript.
Author Response
- The approaches to evaluate the mechanism of action of C60 fullerene in HepG2 cells are clearly insufficient. First of all, if the intention is to evaluate and quantify changes in the expression of proteins in treated HepG2 cells, a much more direct and easy approach is the western blot. In addition, the quality of the immunocytochemical analyses performed in these cells is poor.
Response: The main aim of our study was to assess C60 fullerene ability to treat HCC in vivo (i.e. its impact on liver morphology, liver function blood serum markers, animal survival and metastasis). To evaluate the mechanisms of C60 antitumor action we further assessed liver redox state (as C60 fullerene is known as powerful free radical scavenger), and then used HepG2 cells to discover if C60 fullerene directly affects tumor cell growth, and if so – how. Thus, HepG2 cell investigations were additional and helped us to explain in vivo findings. We quite agree with the reviewer that Western blot is essential for accurate quantitative analysis of proteins expression. Unfortunately, we had no ability to use Western blot (lack of equipment), therefore, we used IHC and appropriate quantitative approaches instead. Probably, IHC pictures don’t provide so clear evidence as Western blot gels do, however, it should be noted than HepG2 cells poorly expressed both p53 and vimentin.
- Moreover, to properly study this issue, authors should have performed analysis to evaluate the migratory capacity of the cells as well as the effects of C60 fullerene in the induction of apoptosis or the reduction of cell proliferation. The analysis of the expression of two proteins in one cell line can not account for the conclusions extracted from these experiments.
Response: We didn’t aim to investigate cell migratory capacity in vitro, because we observed inhibition of metastasis caused by C60FAS in vivo. Moreover, our finding about C60 fullerene impact on vimentin expression was quite surprising, and we tried our best to explain it in Discussion section. Analysis of C60 fullerene effect on the induction of cell apoptosis was performed, appropriate issues were added to Materials and Methods (2.2.1) and Results (3.2) sections, additionally, Figure 2 was modified.
- More importantly, one fails to understand the rationale of the conclusions achieved by the authors in the discussion section (lines 351-362). The authors linked an increase of vimentin expression in HepG2 cells (a protein which has been extensively characterized as a protein involved in the EMT process that results in the increment of the migratory and invasive capacities of the cells, which finally lead to the metastatic spreading of tumors) with the resolution of fibrosis in their animals by metalloproteinase action, and that is without measuring anything related to these proteins (i.e. activity and expression) in any of the models used in this study.
Response: We studied the expression of vimentin as a protein highly expressed in activated hepatic stellate cells and expected its inhibition by C60FAS. Our finding was quite surprising (we performed in vitro tests 3 times in triplicate). As we observed substantial anti-fibrotic, antitumor and antimetastatic effect of C60 fullerene under HCC model in vivo, we suggested that C60FAS (at least if applied in tested dose due to tested term) unlikely induced EMT and HSC activation in vivo. So, we tried our best to explain the obtained results in Discussion. Contradiction of obtained in vitro results with literature data might be explained by different experimental conditions and requires further investigation (which, however, is out of the aim of current study).
- It is interesting, as well as unsettling, that around 40% of the rats die after few weeks (3-4 weeks according to Figure 3) of treatment, when advanced fibrosis and cirrhosis is achieved 8-12 weeks after starting the treatment. What is the explanation for this?
Response: Presented survival curves demonstrated post-treatment survival, i.e. survival after the interventions were terminated. “0 week” time point means not start of the experiment, but start of post-treatment study (corresponded to the 22nd week of experiment), whereas “the 32nd week” time point means the 32nd week of post-treatment study (corresponded to the 54th week of experiment). That’s why we observed animals’ death at so “early” stages (corresponded to 25-27 weeks from the start of experiment and well-developed HCC with metastasis). Appropriate clarification was added to Figure legend.
- It is also curious that authors choose 5-FU as standard drug to compare the efficacy of C60 fullerene when the current first line approved treatment for HCC is the tyrosine kinase inhibitor sorafenib, and more recently the focus of attention in this area is the application of immunotherapy. What is the explanation for this?
Response: We agree with the reviewer, that the only drug approved by US FDA as first-line treatment of HCC is sorafenib. However, despite underestimated role of chemotherapy in HCC treatment, chemotherapy regimens which include 5FU are widely used against advanced HCC as a second-line treatment (doi: 10.2147/JHC.S124366; doi: 10.3892/ol.2018.8242). Furthermore, as we investigated the substance with wide range of cellular targets (C60 fullerene), we chose as a reference another substance with no targeted action (unlike sorafenib - Raf kinase, VEGFRs 1-3 and the PDGFR-β inhibitor). Moreover, 5FU is widely used itself and as a reference in studies dedicated to HCC mechanisms and treatment (doi:10.1186/s13046-016-0349-4; doi:10.4149/neo_2017_304). That is why we proposed 5FU as an adequate reference for assessment of C60 fullerene efficacy against HCC.
- Another overachieving explanation for the results of this study is found in discussion section (lines 394-405). Authors stablished a link around NRF2 based on a couple of references, generating even Figure 5 to explain these interactions. However, NRF2 or anything related with this protein is measured or evaluated in any of the models used in this study. Why authors did not determine the expression/localization of NRF2 in HepG2 cells of livers from rats by the same immunocytochemical assays showed in Figure 2?
Response: We agree with the reviewer that we took an explanation the obtained results too far. Excessive speculations with no direct relation to the obtained results were removed from the Discussion.
- Another inexplicable sentence is found in lines 448-450: “Therefore, we could suggest that the mechanisms of C60 fullerene action are complex, and this fact, in our opinion, explains its significant anti-inflammatory and anti-fibrotic effect”. What is the meaning of this?
Response: We meant that multiplicity of the mechanisms of C60 fullerene action could contribute to its therapeutic effect because of multiplicity of cellular targets and, therefore, ways of action. The text was modified accordingly.
- Finally, the manuscript should be corrected by an English native speaker since multiple grammar errors are found through the manuscript.
Response: We checked the manuscript thoroughly and fixed errors.
Reviewer 2 Report
The manuscript provides important data and valuable results. The conclusions are fully justified. However, there are a few comments to be considered.
It is known that aggregation and stability of C60 fullerene particles in water affect bioavailability and toxicity to an organism. Levels of aggregation and, for example, zeta potential are not presented and discussed. It is difficult to generalize results without these factors.
In Design of the study, the experimental periods are described, but it is not easy to follow it. I would recommend to insert an outline (figure) of experimental design with in vivo treatments and periods.
Table 1 demonstrates measurements at 0, 10 and 100 mg/ml of C60FAS. There is no explanation why 10 and 100 levels were chosen.
The results (and discussion) often indicate how many times values were different, however, it is difficult to find out which treatments are compared. The text needs to be modified.
minor
Abstract: Avoid abbreviations when possible.
Main text: there are abbreviations not explained. Probably clear for experts in the field but not for broader readers. Abbreviations used in tables ought to be explained likely in the list of abbreviations. For example, explain: IC50, SFU IC50, ARE, EGFR, FGFR.
L371: (Table 2) comparing HCC+C60FAS to HCC, and to the control, there were no significant increases. Further, CAT, GST, GP and GSH vs HCC, and vs control did not decrease.
Author Response
- It is known that aggregation and stability of C60 fullerene particles in water affect bioavailability and toxicity to an organism. Levels of aggregation and, for example, zeta potential are not presented and discussed. It is difficult to generalize results without these factors.
Response: Indeed, the aggregation and stability of C60 fullerene particles in water affect bioavailability and toxicity to an organism. Therefore, we conducted the dynamic light scattering (DLS) and zeta potential measurements for ascertaining the hydrodynamic size and electrokinetic potential of the prepared C60FAS. Appropriate issues were added to Materials and Methods (2.1) and Results (3.1) sections.
- In Design of the study, the experimental periods are described, but it is not easy to follow it. I would recommend to insert an outline (figure) of experimental design with in vivo treatments and periods.
Response: We added a scheme of experiment (Figure 1).
- Table 1 demonstrates measurements at 0, 10 and 100 mg/ml of C60 There is no explanation why 10 and 100 levels were chosen.
Response: Concentrations of C60 fullerene were chosen as those demonstrating substantial effect on cell growth and apoptosis. We tried to assess if the effects of C60 fullerene on tested parameters could contribute to its toxicity. Appropriate explanation was added to the text (Materials and Methods 2.2.2)
- The results (and discussion) often indicate how many times values were different, however, it is difficult to find out which treatments are compared. The text needs to be modified.
Response: We modified the text accordingly to avoid misunderstanding.
- Abstract: Avoid abbreviations when possible.
Response: We excluded abbreviations in Abstract whenever possible. The only one left is HCC (hepatocellular carcinoma). It cannot be avoided because of need to use many times.
- Main text: there are abbreviations not explained. Probably clear for experts in the field but not for broader readers. Abbreviations used in tables ought to be explained likely in the list of abbreviations. For example, explain: IC50, SFU IC50, ARE, EGFR, FGFR.
Response: We checked the text of the manuscript thoroughly and added all missed explanations.
- L371: (Table 2) comparing HCC+C60FAS to HCC, and to the control, there were no significant increases. Further, CAT, GST, GP and GSH vs HCC, and vs control did not decrease.
Response: In this paragraph (L371) we meant changes in HepG2 cells (presented in Table 1). Table 3 demonstrated biochemical parameters of livers of HCC-rats received different treatments. This misunderstanding likely occurred through lack of appropriate indication of compared groups, as the reviewer mentioned in point 4. We checked the tables and description thoroughly and modified the text accordingly to avoid misunderstanding.
Reviewer 3 Report
Submitted article by Kuznietsova and co-workers, describes the use of [60]fullerene nanoparticles in the treatment of hepatocellular carcinoma (HCC). The authors try to figure out the anti-oxidant properties of fullerene nanomaterials , in particular the effects on p53, vimentin and G6PD biomarkers. What is more important, the manuscript shows in vivo results, reporting promising effects of C60FAS on liver enzymes and tumor metastasis. Although manuscript seems to be interesting, some issues must be taken into consideration:
- Additional tests for determination of cytotoxicity (i.e. PI staining and flow cytometry) should be performed. MTT shows only effects on mitochondrial enzymes and could be false.
- It’s not clear why the authors used sophisticated methods for studing effects of C60FAS on three proteins: p53, vimentin and G6PD. Additional studies have to be performed using Western-Blot technique
- The authors have studied the effects of C60FAS on cellular redox state markers (i.e. MDA, PCG, GSH). They used spectrofluorimetric assays to determine if desirable enzyme works. But the C60FAS has a strong absorption at 340 nm- so part of the results may be false due to absorption of fullerene nanomaterial. I suggest another method for determination of protein upregulation/downregulation, like Western Blot technique.
- In terms of animal studies, there is no information about sex and the age of animals. What volumes of DEN and CCl4 were injected to the rats? What is a total dose of C60FAS that was used per animal? Additionally I’ve got concerns about bioethics of the study. The animal were “active” in the study till the “natural” death. Why after reaching a terminal point, the were not sacrificed?
In conclusion, I suggest the publication of the submitted manuscript in Pharmaceutics but after major revision.
Author Response
- Additional tests for determination of cytotoxicity (i.e. PI staining and flow cytometry) should be performed. MTT shows only effects on mitochondrial enzymes and could be false.
Response: We added an assessment of C60FAS proapoptotic action (Annexin V/Propidium Iodide assay). Appropriate issues were added to Materials and Methods (2.2.1) and Results (3.2) sections, additionally, Figure 2 was modified.
- It’s not clear why the authors used sophisticated methods for studing effects of C60FAS on three proteins: p53, vimentin and G6PD. Additional studies have to be performed using Western-Blot technique
Response: We aimed to assess the C60 fullerene impact on activity of G6PD as the key rate-controlling enzyme of the pentose-phosphate pathway, that is why we thought the spectrophotometric estimation of the enzyme’s activity would be enough to achieve our goal. To discover the mechanisms of C60FAS antitumor action (observed in vivo), we assessed C60 fullerene impact on cell survival (MTT assay) and on expression of one of the main proteins responsible for tumor cell death – p53. Then, to discover the possible mechanisms of C60 fullerene anti-fibrotic action (observed in vivo), we investigated its impact on expression of vimentin – one of the markers of EMT and hepatic stellate cells’ activation (our results about vimentin were surprising and we tried our best to explain them in Discussion). We quite agree with the reviewer that Western blot is essential for accurate quantitative assessment of protein expression. Unfortunately, we had no ability to use Western blot (lack of equipment), therefore, we used IHC and appropriate quantitative approaches instead.
- The authors have studied the effects of C60FAS on cellular redox state markers (i.e. MDA, PCG, GSH). They used spectrofluorimetric assays to determine if desirable enzyme works. But the C60FAS has a strong absorption at 340 nm- so part of the results may be false due to absorption of fullerene nanomaterial. I suggest another method for determination of protein upregulation/downregulation, like Western Blot technique.
Response: For sure, C60FAS has a strong absorption at the mentioned wavelength. However, we didn’t use the material where high amounts of C60 fullerene (high enough to affect the absorption) may be presented. Thus, we used tissue samples extracted in 24 h after the interventions were terminated (see Materials and Methods 2.3.1) and washed with saline etc. to avoid impact of the blood contents (including C60 fullerene which hadn’t been excreted at the time of the sacrifice) (see 2.3.5). If using HepG2 cells for biochemical studies, we removed cell culture medium contained C60 fullerene and washed cells with fresh one before cells were frozen (appropriate clarification was added to the test of the manuscript – Materials and Methods 2.2.3). So, we hope we excluded C60 fullerene from the analyzed material enough at least to avoid its impact on sample absorption at the 340 nm.
- In terms of animal studies, there is no information about sex and the age of animals. What volumes of DEN and CCl4 were injected to the rats? What is a total dose of C60FAS that was used per animal? Additionally I’ve got concerns about bioethics of the study. The animal were “active” in the study till the “natural” death. Why after reaching a terminal point, the were not sacrificed?
Response: We indicated the sex of the animals (Materials and Methods 2.3 line 140) and their initial body weight because of choosing the weight as a criterion for including into the study. Strong dependence of weight from the age of healthy animals if maintained at the same condition is obvious. That is why we didn’t indicate the initial age of the tested animals in the study design description. All the animals included into the study were the same age (1 month), and we included these data into the study design description in revised manuscript (Materials and Methods 2.3) to avoid any misunderstanding. The total volumes of DEN, CCl4, 5FU solutions and C60FAS injected per rat were also added. Describing the design of the study, we indicated that the animals were treated for 22 weeks, then the halves of the animals from each group (chosen randomly) were sacrificed (Materials and Methods 2.3.1), another halves were left without any treatment for survival assay for the next 32 weeks (Materials and Methods 2.3.1). After the reaching this time point (chosen as that being enough to estimate post-treatment median survival times and not contradicting DeGeorge JJ, Ahn CH, Andrews PA, et al. Regulatory considerations for preclinical development of anticancer drugs. Cancer Chemother Pharmacol. 1998;41(3):173-185 and ICH M3 (R2) Non-clinical safety studies for the conduct of human clinical trials for pharmaceuticals, adopted 11.02.2013) the animals still alive were sacrificed (Materials and methods 2.3.2) to avoid suffering and in accordance with ICH M3 (R2).
Round 2
Reviewer 1 Report
The authors have answered properly most of the commentaries from the reviewers and have performed several experiments to complement the in vitro studies. Nevertheless, I am still not convinced about the explanation given in the discussion for the results concerning vimentin expression analysis in HepG2 cells.
Author Response
The authors have answered properly most of the commentaries from the reviewers and have performed several experiments to complement the in vitro studies. Nevertheless, I am still not convinced about the explanation given in the discussion for the results concerning vimentin expression analysis in HepG2 cells.
Response: We are very thankful you for a thorough analysis of our work and valuable comments. We improved the explanation of the results concerning vimentin expression in HepG2 cells and tried to avoid unnecessary and unconfirmed speculations (lines 407-423).
Reviewer 2 Report
The authors considered my comments and modified the manuscript. The work is well described and suitable for publishing.
Author Response
The authors considered my comments and modified the manuscript. The work is well described and suitable for publishing.
Response: We are very thankful you for a thorough analysis of our work and valuable comments.
Reviewer 3 Report
The manuscript should be accepted in current form. The authors included additional cytotoxicity studies and updated the methodology part.
Author Response
The manuscript should be accepted in current form. The authors included additional cytotoxicity studies and updated the methodology part.
Response: We are very thankful you for a thorough analysis of our work and valuable comments.